# Temporal Evolution and Regional Properties of Aerosol over the South China Sea

**Jie Chen** [1,2,3], **Wenyue Zhu** [1,3,*], **Qiang Liu** [1,3], **Xianmei Qian** [1,3], **Xiaowei Chen** [1,3], **Jianjie Zheng** [1,3], **Tao Yang** [1,2,3], **Qiuyi Xu** [1,2,3] and **Tengfei Yang** [1,2,3]

1. Key Laboratory of Atmospheric Optics, Anhui Institute of Optics and Fine Mechanics, HFIPS, Chinese Academy of Sciences, Hefei 230031, China
2. Science Island Branch of Graduate School, University of Science and Technology of China, Hefei 230026, China
3. Advanced Laser Technology Laboratory of Anhui Province, Hefei 230037, China
* Correspondence: zhuwenyue@aiofm.ac.cn

**Abstract:** Aerosol robotic network (AERONET) data from Dongsha Island (20.699N, 116.729E) and Taiping Island (35.90N, 3.03W) over the South China Sea (SCS) from January 2018 to December 2020 were used to analyze and discuss the temporal evolution properties of aerosols in the South China Sea. Surrounding AERONET stations (Hong Kong, NSPO, Nha Trang and Singapore) were also used to analyze regional characteristics. High aerosol loads over Dongsha were strongly associated with the anthropogenic fine particle transport from the southeastern coast of China and occasional advection of desert dust from Mongolian areas. The high fine aerosol loading in Taiping originates from the region between Singapore and Indonesia. Compared with other marine islands in the world, SCS was not a pure marine aerosol environment and was affected by terrestrial aerosols. In the Taiping area, aerosol optical depth $\tau$ (500 nm) was $0.17 \pm 0.13$ and the average Ångström exponent $\alpha$ (440–870 nm) was $0.96 \pm 0.36$. However, that of Dongsha shows the larger values of $\tau$ ($0.26 \pm 0.21$) and $\alpha$ ($1.1 \pm 0.38$), indicating that there are large fluctuations in aerosol concentration and size. Aerosol loads in different regions of the SCS due to uneven socioeconomic and complex meteorological systems, such as those of the coastal cities of China, Singapore, and the region between Singapore and Indonesia, contribute to the high optical depth. The special meteorological regime and aerosol source mechanism in the SCS leads to the obvious seasonal cycle of aerosol optical depth and Ångström index. Moreover, the loading variations of aerosols on Dongsha Island and Taiping Island were highly consistent with those of coastal cities around them, suggesting the significant effect of the aerosol in the SCS by the surrounding coastal cities, although the aerosol optical depth in these two places was much lower than that in the surrounding cities.

**Keywords:** South China Sea; AERONET; aerosol optical depth





## 1. Introduction

The South China Sea (SCS) connects the Pacific Ocean with the Indian Sea and is one of the largest marginal seas in Southeast Asia [1–3]. In such a low latitude sea, it is often affected by aerosol eruptions from Asia, resulting in local climate change anomalies [3–5]. The complex physical, biological, geographical, monsoon climate and complicated economy are of great importance in the combined effect of aerosol properties [6]. In particular, the SCS receives substantial amounts of different types of aerosols from surrounding regions due to various economic activity [5,7].

Many efforts have been made by many groups to characterize this area by using two main methods: sampling and remote sensing. The sampling method of atmospheric aerosols is one of the main methods used to research the pollutant sources and related chemical properties in this area. Xiao et al. [8] found that fossil fuel combustion on Yongxing Island (especially coal in the coastal areas of China) was an important source of $NO_3^-$ (56%)

and $SO_4^{2-}$ (22%), and biomass burning accounted for 41% of $K^+$ in Asia. On Dongsha Island, carbonaceous content and water-soluble ions are the dominant components of total suspended particulates (TSP), which are affected by mobile vehicles and coal and biomass burning [9]. Zhang et al. [10] reported that methanesulfonic acid measured at PM2.5 aerosol samples over the northern SCS was comparable to those over other coastal regions. The sampling study mainly focused on the short-term analysis of pollutant composition in this region and its nearby islands, but did not carry out the long-term exploration of aerosol in the South China Sea. In addition, remote sensing provides an important tool for understanding the function of aerosols in Earth's radiation budget. From satellite and ground remote sensing measurements, aerosol column characteristics can be obtained. Smirnov et al. [11,12] conducted a number of studies on aerosol optical properties in various maritime and coastal areas, except the SCS, by using AERONET. Studies gave a generally accepted criteria for the determination of "pure marine aerosol": AOD (500 nm) smaller than 0.15 and Ångström exponent $\alpha$ (440–870 nm) less than 1. Itahashi et al. [13] and Zheng et al. [14] used MODIS to analyse aerosol optical depths in the Seto Inland Sea and East Sea near the SCS. Reid et al. [15] analyzed 7 Southeast Asian Studies (7-SEAS) and found that the strong monsoon effected the transport and removal of aerosol particles by the AERONET and Lidar data in SCS. Many scholars have studied local aerosol characteristics and analysed microphysical properties, optical properties and aerosol distributions based on ground-based measurements surrounding the SCS [16,17].

Although the abovementioned studies focused on the analysis of local pollution and aerosol properties, they only targeted short-term experimental measurements or focused mainly on high aerosol load cases. Therefore, it is necessary to study the aerosol properties over the SCS to evaluate the aerosol regimes, which is significant with regard to researching temporal evolution and regional properties and makes up for the absence of aerosol models over this explored region. In this study, the aerosol loading, source and spatio-temporal variation of Dongsha Island and Taiping Island in the South China Sea from 1 January 2018 to 31 December 2020 were presented by analyzing AERONET data. The optical properties of aerosol over the SCS are analysed and discussed by considering the complexity of aerosol origin and propagation, meteorology and geography, which is of great significance for the study of the regional climate and atmospheric radiation transfer model.

## 2. Data Sources and Research Sites

### 2.1. AERONET Measurements

As a standard automatic solar photometer, a CE-318 was used to measure the characteristics of columnar aerosols in the AERONET network [18]. The full viewing angle of the instrument was 1.2°, and direct sunlight measurement was conducted at 340, 380, 440, 500, 670, 870, 940 and 1020 nm (nominal wavelength). Moreover, the $\tau$ at each wavelength was retrieved from the direct sunlight measurements, except $\tau$ (940 nm), since it was used to calculate water vapour absorption. A further description about the photometer can be found in Holben et al. [18]. The AERONET collaboration provides spectral aerosol optical depth (AOD), retrieval products, and global distribution observations of precipitable water under different aerosol states. In order to understand the AOD corresponding to short wavelength and long wavelength, the aerosol optical properties were studied based on the wavelengths of 380, 500 and 1020 nm. In addition, the Ångström exponent $\alpha$ (440–870 nm) can effectively show the variation of particle size [19]. Ångström exponent $\alpha$ was obtained from 440 nm to 870 nm by exponential fitting of the aerosol optical depth at different wavelengths. Fine aerosol optical depth at 500 nm ($\tau^F$ (500 nm)), coarse aerosol optical depth at 500 nm ($\tau^C$ (500 nm)), and fraction of optical depth of fine mode particles (FMF) were determined by de-convolution algorithm [20]. Currently, AERONET contains three levels of data, among which level 1 data was unfiltered, level 1.5 data was quality controlled and filtered, and the level 2 data had quality assurance [21].

## 2.2. Sites Selection

The location of each AERONET site is shown in Figure 1. This study focused on the AERONET stations at Dongsha Island (20.699N, 116.729E) and Taiping Island (35.90N, 3.03W) over the SCS. Dongsha Island lies in the north of the SCS and is surrounded by the Philippines, southern China and the Indochinese Peninsula in the distance (Figure 1). Taiping Island is located in the northwest of the Nansha Islands, approximately 740 km away from the Xisha Islands, and only 1000 km away from Singapore at the eastern mouth of the Strait of Malacca. It is valuable in the aspects of channel safety, shipwreck notification, meteorological monitoring, international aviation information, etc. NSPO (24.784N, 121.001E) of AERONET station, located approximately 628 km northeast of Dongsha Island, is the abbreviation of the Taiwan Space Center. It was used to evaluate the impact of urban sources of aerosols on Dongsha Island. The Hongkong station (22.483N, 114.17E) and Nha Trang station (12.205N, 109.206E) are located 340 km and 620 km away from Dongsha Island and Taiping Island, respectively, to assess the spatial distribution of aerosols. The Singapore station (1.298N, 103.780E) is located 1000 km southwest of Taiping Island to assess the source of aerosol long-distance transport. Since the data from AERONET aerosol sites were not continuous and synchronous in time, in order to obtain the recent aerosol optical depth, seasonal and spatial distribution characteristics in the South China Sea, our study chose 2018 to 2020 as the research period.

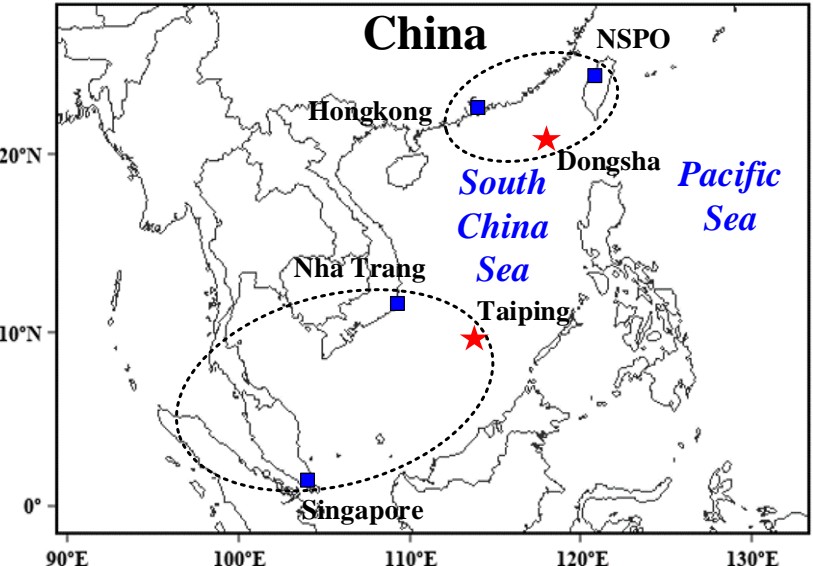

**Figure 1.** Locations of the observation sites in this study. Dongsha Island and Taiping Island sites are shown as red stars, and other sites are shown as blue squares. All stations are affiliated with AERONET.

## 2.3. EAC4 and ERA5

EAC4 (ECMWF Atmospheric Composition Reanalysis 4) is the fourth generation European Centre for Medium-Range Weather Forecasts (ECWMF) global atmospheric composition reanalysis. The reanalysis uses atmospheric models based on physical and chemical laws to combine the model data with observation data from all over the world to form a global complete and consistent dataset. This principle, known as data assimilation, is based on the method used by the numerical weather prediction center and the air quality prediction center. MODIS observations of AOD (550 nm) over ocean and land (except bright surfaces) was also assimilated in EAC. Tropospheric aerosols were classified into five types: sea salt, organic matter, desert dust, sulfate aerosols and black carbon. All aerosol species are regarded as tracers, and are incorporated into the vertical diffusion and convection scheme of the integrated forecasting system (IFS) [22]. Detailed methods for EAC4 to obtain various aerosol optical depths can be found in many studies [23,24]. The

AOD and the mass concentration in Dongsha and Taiping were obtained through EAC4 in this study [25]. In addition, the high consistency ($R^2$ = 0.92) of AOD obtained by EAC and AERONET indicates that the data of EAC4 can be used to effectively verify the analysis results (Figure 2).

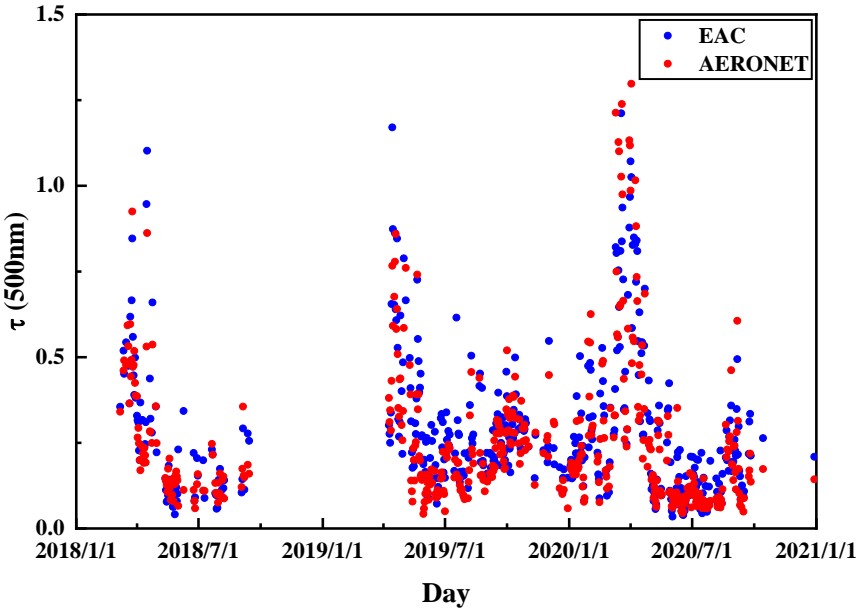

**Figure 2.** Comparison of $\tau$ (500 nm) obtained by AERONET and EAC4 at Dongsha Island.

ERA5 is the ECMWF's fifth generation reanalysis of global climate and weather over the past 40 to 70 years. Hourly estimates of large amounts of atmosphere, ocean waves and land surfaces was provided by ERA5 [26]. In this study, the wind speed, wind direction and wave height of 10 metres were obtained to access meteorological characteristics of the South China Sea.

### 2.4. Air Mass Trajectories

HYSPLIT is a complex and widely used system that uses the Lagrangian approach and Eulerian methodology for the calculation of air encapsulation paths, complex transport, pollutant dispersion, chemical transformation, and particle deposition simulations [27]. By calculating five-day backwards trajectories ending at 00:00 UTC at Dongsha Island and Taiping Island for 500 m above ground level using the HYSPLIT model every day for three years, the source and transport of air mass to our target AERONET sites were characterized. We chose five days as the compute cycles for this backwards trajectory analysis to ensure trajectory integrity and reduce computational burden. In addition, the analysis height of 500 metres can effectively cover multiple air mass sources and transport trajectories at other altitudes that are similar to this height.

### 3. Results and Analysis

#### 3.1. Time Evolution of Aerosol Properties over Dongsha Island and Taiping Island

Figure 3(a$_1$,b$_1$) shows the time evolution of daily mean values of $\tau$ (380, 500 and 1020 nm) and $\alpha$ (440–870 nm) measured at Dongsha Island in the SCS from 1 January 2018 to 31 December 2020. Much of the missing data in the $\tau$ ($\lambda$) and $\alpha$ (440–870 nm) series is due to invalid data. All of the daily average values of Dongsha Island are presented in Table 1. A distinct feature is the large variability of $\tau$ ($\lambda$) ($\tau$ (500 nm) that ranged from 0.04 (Min) to 1.3 (Max), which is closely linked to the diversity of air mass sources in the study area, as mentioned below. In order to evaluate the magnitude of variation in the various data sets, the coefficient of fluctuation (FOC) was expressed by dividing the standard deviation of the data set by the mean. As shown in Table 1, $\tau$ ($\lambda$) at 1020 nm showed smaller FOC than

that at 380 nm. It is well known that $\tau(\lambda)$ is more susceptible to fine particles (radius below 0.5 μm) at short wavelengths, whereas it is more sensitive to coarse particles (radius above 0.5 μm) at long wavelengths [28]. In accordance with larger FOC of $\tau^F$ (500 nm) compared to $\tau^C$ (500 nm) (Table 1), the higher variability of $\tau(\lambda)$ for shorter wavelengths indicates strong variability in the fine particle load over Dongsha Island. In this region, the large $\tau(\lambda)$ fluctuations corresponding to long wavelengths may be caused by the intrusion of marine aerosols and dust, as described below. In addition, coarse particles in the atmosphere have a shorter lifetime than fine particles, which may also be a reason for the large coarse particle fluctuations $\tau^C(\lambda)$. The $\alpha$ (440–870 nm) also showed great variability, ranging from 0.15 to 1.95 with an average of $1.1 \pm 0.38$, indicating that the aerosol types (coarse particles, fine particles and different mixtures of coarse particles and fine particles) were different under different atmospheric conditions. Notably, $\alpha$ (400–870 nm) was greater than 1 in 66% of the analyzed days, indicating that the aerosol population of Dongsha Island was dominated by fine particles in most of the analyzed days. This conclusion was further supported by the analysis of fine mode fraction (FMF), which ranged from 0.18 to 0.97 (mean $0.65 \pm 0.19$), with 78% of the days analyzed having a daily mean greater than 0.5.

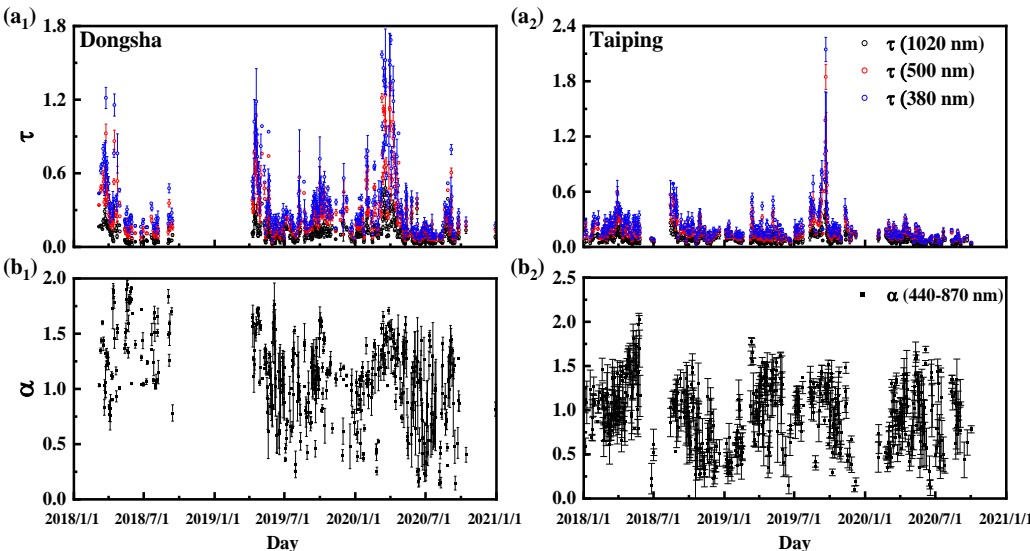

**Figure 3.** (**a**) Daily average aerosol optical depth at 380, 500 and 1020 nm and (**b**) Ångström index calculated from 440 to 870 nm at Dongsha Island ($a_1,b_1$) and Taiping Island ($a_2,b_2$) in the South China Sea from 1 January 2018 to 31 December 2020. The error bars are the standard deviations.

**Table 1.** Multi-wavelength (1020, 500 and 380 nm) aerosol optical depth $\tau$, Ångström exponent ($\alpha$ (440–870)), fine aerosol optical depths at 500 nm ($\tau^F$ (500 nm), coarse aerosol optical depths at $\tau^C$ (500 nm)) and fraction of optical depth of fine mode particles (FMF) in Dongsha from 1 January 2018 to 31 December 2020. All data were processed using daily averages. The total number of samples (N total), average coefficient (Mean), standard deviation (SD), Minimum (Min), Median (Med), Maximum (Max) and the coefficient of fluctuation (FOC) are also counted to assess variations in the data.

| | N Total | Mean | SD | Min | Med | Max | FOC |
|---|---|---|---|---|---|---|---|
| $\tau$ (1020 nm) | 456 | 0.11 | 0.08 | 0.02 | 0.1 | 0.48 | 69% |
| $\tau$ (500 nm) | 456 | 0.26 | 0.21 | 0.04 | 0.19 | 1.3 | 84% |
| $\tau$ (380 nm) | 456 | 0.34 | 0.28 | 0.05 | 0.25 | 1.69 | 82% |
| $\alpha$ | 456 | 1.1 | 0.38 | 0.15 | 1.13 | 1.952 | 34% |
| $\tau^F$ | 453 | 0.18 | 0.21 | 0.01 | 0.111 | 1.25 | 111% |
| $\tau^C$ | 453 | 0.06 | 0.04 | 0.01 | 0.06 | 0.34 | 64% |
| FMF | 453 | 0.65 | 0.19 | 0.18 | 0.66 | 0.97 | 30% |

Eight islands around the world are shown in Figure 4 to compare aerosol loads. These islands or AERONET stations cover the Pacific (American_Samoa, 14.247S, 170.564W; Okinawa_Hedo, 26.867N, 128.249E), Indian (Amsterdam_Island, 37.800S, 77.572E), Atlantic (ARM_Graciosa, 39.091N, 28.029W; Ascension_Island, 7.976S, 14.415W), Mediterranean (Lampedusa, 35.517N, 12.632E) and SCS (Dongsha and Taiping Islands). The mean $\tau$ (500 nm) values over the South China Sea are significantly higher than those over the open oceanic areas (American Samoa, Amsterdam Island, and ARM_Graciosa) without the influence of long-distance transport. Additionally, the $\alpha$ (440–870 nm) and FMF values obtained in this study were smaller than those of open oceanic areas for maritime aerosols. Later in the article, we compare the Dongsha observations with those of three nearby AERONET sites during the same period. Pure maritime situations can be generally found when $\tau$ (500 nm) < 0.15 and $\alpha$ (440–870 nm) < 1 according to Smirnov et al. [12]. Based on this evaluation criterion, 16 percent of the analyzed days on Dongsha Island were pure sea conditions, the least of all the islands. Clean maritime conditions observed over Taiping Island were more frequent than those observed over Dongsha Island and were still far less frequent than in open oceanic areas. In addition, Okanaw_Hebe, Lampedusa and Ascension_Island show similar performance in marine conditions. Marine areas that are mostly surrounded by the Baltic, the Mediterranean and Sea of Japan were excluded from the pure maritime research [29]. The SCS is a marginal sea in southern China surrounded by coastal cities and regions in southeast China, the Philippines, the Greater Sunda Islands and the Indochina Peninsula. Dongsha Island and Taiping Island are not directly affected by human activities, but air masses reaching the two islands may be influenced by anthropogenic aerosols during their passage over the SCS and continents. Figure 3 (a$_2$,b$_2$) shows the aerosol properties in Taiping. The mean $\tau$ (500 nm) value of 0.17 and $\alpha$ (440–870 nm) value of 0.96 over Taiping Island were significantly lower than the mean $\tau$ (500 nm) value of 0.26 and $\alpha$ (440–870 nm) value of 1.2 over Dongsha Island. Additionally, the average FMF value of Taiping Island was lower than that of Dongsha Island (Table 2). The difference in the presence of aerosols at these two sites can be explained by their different locations. Taiping is farther away from land than Dongsha, and the surrounding air mass will carry more sea salt aerosol particles, increase the proportion of coarse particles, and reduce the optical depth.

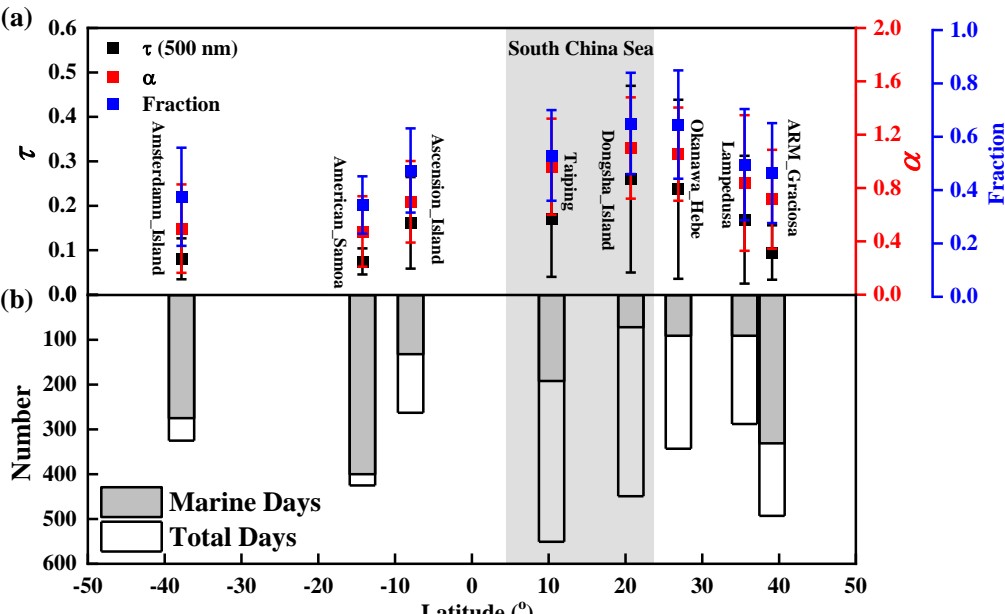

**Figure 4.** Global annual average, meridional distributions of (**a**) $\tau$ at 500 nm, $\alpha$ (440–870 nm), FMF; (**b**) the total number of samples and pure maritime days.

**Table 2.** Multi-wavelength (1020, 500 and 380 nm) aerosol optical depth $\tau$, Ångström exponent ($\alpha$ (440–870)), fine aerosol optical depths at 500 nm ($\tau^F$ (500 nm), coarse aerosol optical depths at $\tau^C$ (500 nm)) and fraction of optical depth of fine mode particles (FMF) in Taiping from 1 January 2018 to 31 December 2020; All data were processed using daily averages. The total number of samples (N total), average coefficient (Mean), standard deviation (SD), Minimum (Min), Median (Med), Maximum (Max) and the coefficient of fluctuation (FOC) are also counted to assess variations in the data.

| | N Total | Mean | SD | Min | Med | Max | FOC |
|---|---|---|---|---|---|---|---|
| $\tau$ (1020 nm) | 551 | 0.09 | 0.06 | 0.01 | 0.08 | 0.7 | 65% |
| $\tau$ (500 nm) | 551 | 0.17 | 0.13 | 0.04 | 0.13 | 1.85 | 79% |
| $\tau$ (380 nm) | 551 | 0.21 | 0.16 | 0.04 | 0.17 | 2.14 | 78% |
| $\alpha$ | 551 | 0.96 | 0.36 | 0.11 | 0.98 | 2.03 | 37% |
| $\tau^F$ | 450 | 0.09 | 0.13 | 0.01 | 0.06 | 1.8 | 146% |
| $\tau^C$ | 450 | 0.06 | 0.03 | 0.01 | 0.06 | 0.24 | 54% |
| FMF | 450 | 0.53 | 0.17 | 0.16 | 0.53 | 0.982 | 32% |

As shown in Figure 3($a_1$,$b_1$), Dongsha Island is strongly affected by aerosols for several days each year, with $\tau$ (500 nm) values exceeding 0.35. High aerosol loads ($\tau$ (500 nm) > 0.35) [28] over Dongsha Island were observed for 90 out of 449 days of analysis. These events were observed on 66 analysis days from March to April. In these cases, the 60-day $\alpha$ (440–870 nm) value was higher than 1.1, and the FMF value was higher than 0.7, indicating the dominant role of fine particles. In the remaining high aerosol loading events, high aerosol loading was associated with relatively low $\alpha$ (440–870 nm) values, reaching the lowest $\alpha$ (440–870 nm) values (about 0.83). During these days, the FMF values were also low and reached the lowest mean daily value of 0.52. This behavior indicates that coarse particles mainly transported from other regions, except for sea salt aerosol, as there is no local coarse particle activity in Dongsha. According to the analyses of back trajectories during high aerosol loading days (Figure 5), all of these cases were linked to anthropogenic and dust intrusion from the southeastern coast of China and Mongolia. The important thing to note about these events is that $\tau^F$ (500 nm) was also relatively high and ranged from 0.17 to 1.25 with a mean value of 0.56 ± 0.26. These results highlight the significant contribution of fine model particles, with FMF ranging from 52% to 97% during these high aerosol loading events. A backward trajectory analysis of high aerosol loads on these days shows that the air mass that originated over the Taiwan Strait (57%) and Mongolia (17.12%) in the study area was at a lower altitude (500 m). However, compared with Dongsha Island, high aerosol loads ($\tau$ (500 nm) > 0.35) over Taiping Island can still be observed on 19 of the 450 analysed days, and occurred from August to September, especially in 2018 and 2019. From 10 August to 30 September, the back trajectories in Taiping Island were almost related to high aerosol loads from the region between Singapore and Indonesia. On these days, high aerosol loads were associated with relatively high $\alpha$ (440–870 nm) values, which peaked at about (2.14).

The values of $\tau^F$ (500 nm) were also high (>0.19), and the highest value was 1.8. Since there is no significant anthropogenic activity in Taiping, it is likely that the fine particulate matter in these cases originated mainly from mainland industrial/urban areas. Similarly, the $\alpha$ (440–870 nm) value of Taiping Island was less than 0.9 on two days of high aerosol loading days. However, $\alpha$ (440–870 nm) values were relatively low (<0.88) with high $\tau^F$ (500 nm) values (approximately 0.22). These behaviors suggest that coarse particles were transported from other locations, as there was no local coarse particle activity in Taiping in addition to sea salt aerosols. In this event, the high $\tau$ values observed were associated with a sustained monsoon climate centered in the SCS, which favours the transport of anthropogenic particles emitted from Southeast Asia to the SCS. According to the EAC, the aerosol optical depths of black carbon aerosol (BC), dust aerosol (DU), organic matter aerosol (OM), sulfate aerosol (SU) and sea salt aerosol (SS) in Dongsha and Taiping are

shown in Figure 6. It reveals that high aerosol loading in Dongsha and Taiping Island was caused by the sharp increase in black carbon aerosols, dust aerosols and organic aerosols in addition to sea salt aerosols. Figure 7 shows the distribution of BC in Dongsha and OM in Taiping and its surrounding areas before and after the high aerosol loading in 2019. BC and OM are typical urban aerosols. The appearance of these two aerosols confirms the intrusion from foreign air masses in the process of high aerosol loading. Therefore, the high loading of fine particles in Dongsha mainly originated from BC and OM brought from the southeastern coast of China. A small amount of coarse DU particles were loaded from Mongolia. In addition, the high fine particle loading in Taiping was mainly produced by OM and BC, which came from Southeast Asia, including Singapore and the region between Singapore and Indonesia.

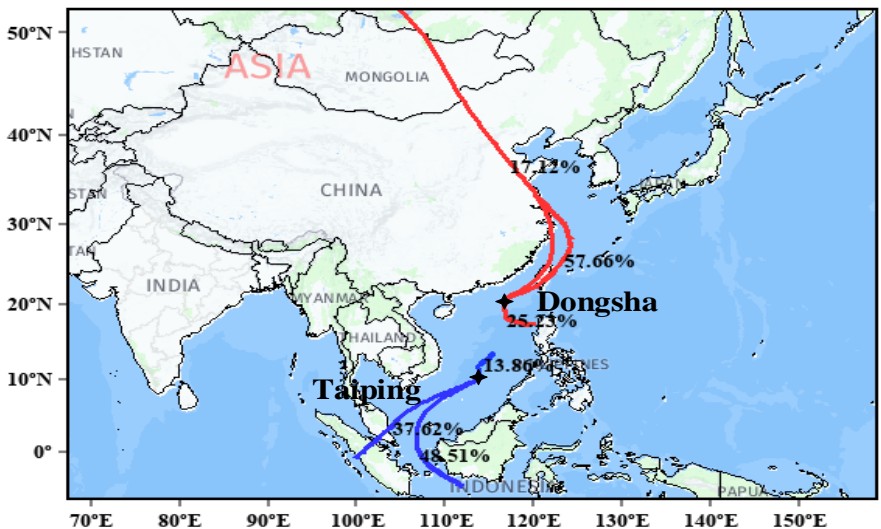

**Figure 5.** Five-day trajectories ending at Dongsha (15 March to 20 April every year) and Taiping (10 August to 30 September every year) at 500 m during high aerosol loading.

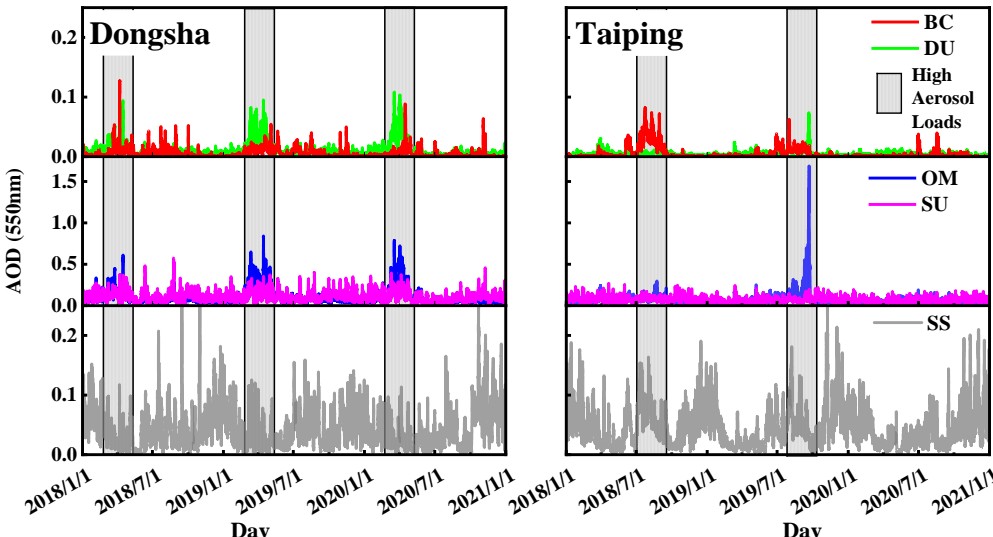

**Figure 6.** Time evolution of AOD at 550 nm of BC, DU, OM, SU and SS obtained from EAC.

*3.2. Back Trajectories and Aerosol Optical Properties over Dongsha Island and Taiping Island*

The Hysplit Dispersion Model was used to calculate the 120-h backward trajectory ending at Dongsha and Taiping to determine the source of the air mass (Figure 8). Each trajectory was linked to the corresponding aerosol optical depths. Previously, this method

has been widely used to evaluate the distribution characteristics of aerosol optical depths of different air masses [30,31].

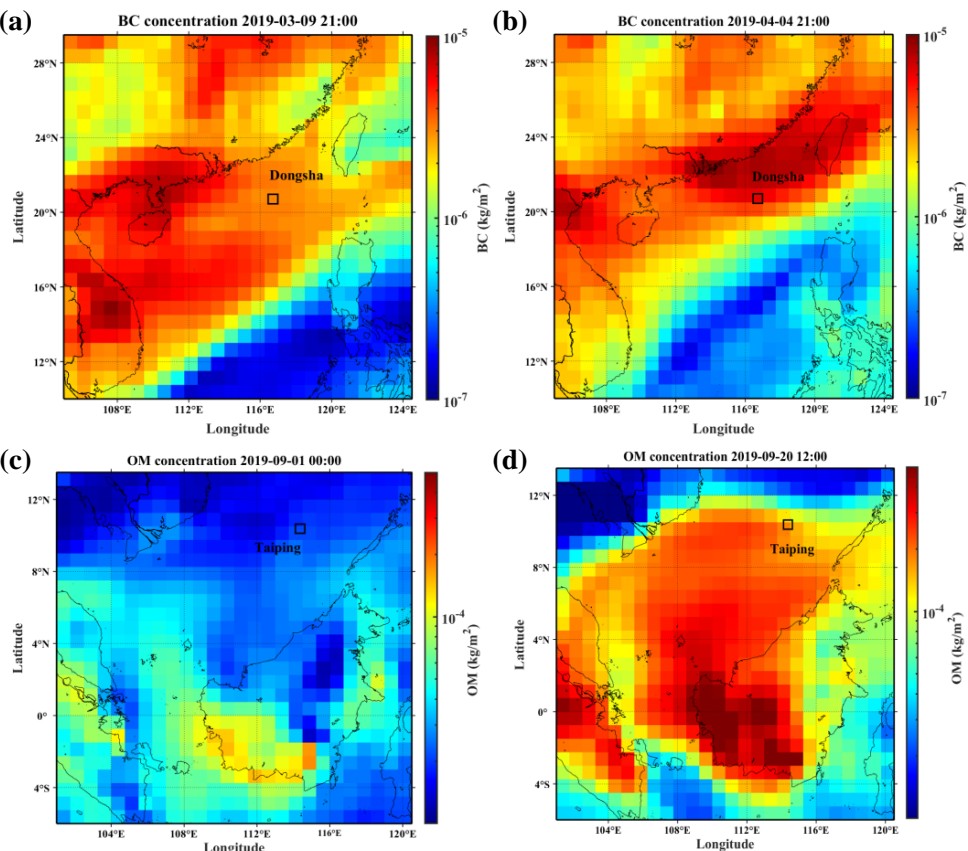

**Figure 7.** Before (**a**) and after (**b**) the sharp increase in black carbon aerosol loading in the Dongsha area; before (**c**) and after (**d**) the sharp increase in organic matter aerosol loading in the Taiping area.

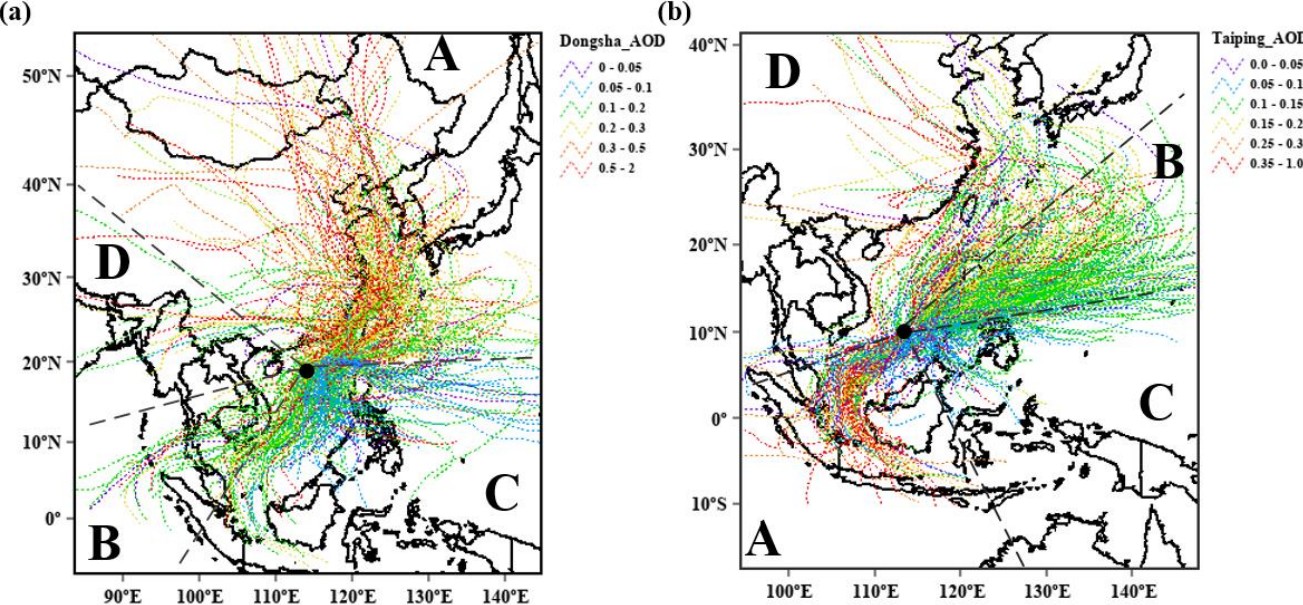

**Figure 8.** Five-day trajectories ending at 500 m from January 2018 to January 2021 at (**a**) Dongsha and (**b**) Taiping; The color of the trajectory represents the value of aerosol optical depth. One trajectory per day was displayed; The different source areas are separated into four sections (A, B, C and D), as shown in the panel.

Broad geographic areas are divided based on the possibility of determining the major sources of aerosol type and optical depth. In this way, the definitions of the A, B, C and D sectors are constrained by optical depth in the different areas. The large degree of arbitrariness may stem from the choice of borders. By considering $\tau$ and the different sectors along the trajectory, the sector of origin of the aerosol can be determined. We defined four broad geographical sectors, displayed in Figure 8, in relation to different aerosol optical depths; four broad geographical sectors are defined in Figure 8. The identified sectors at Dongsha were: (A) North of Dongsha Island, where aerosols mainly come from the southeast coast of China (46%); (B) The western sector (36%), which includes the China-Indochina Peninsula and the SCS; in this region, the prevalent sources produce continental and, to a lesser extent, marine aerosols; (C) the southeast sector (17%), coinciding with the Luzon Strait and the SCS; (D) other parts had no obvious boundary (1%), and the proportion was negligible. The identified sectors at Taiping were: (A) the southwestern portion of Taiping Island (21%), which is largely bounded by Singapore and Indonesia, together which provides an important source of urban aerosols; (B) an east-north sector (55%), coinciding with the Philippines; in this region, the prevalent sources produce continental aerosols and, to a lesser extent, marine aerosols; (C) an East-South sector (10%), which included the Sulu Sea; (D) Other parts (14%), which accounted for less and have unclear boundaries, were not analyzed in this study.

The source distribution of the air mass in the Dongsha area will be described first. The high aerosol loads were well identified from the trajectory analysis and correspond to class A. As we mentioned earlier, aerosols from sector A had the largest average aerosol optical depth ($0.35 \pm 0.23$) (Figure 9($a_1$)). In addition, 80% of $\alpha$ values in class A were higher than 1 (Figure 9($b_1$)). Sector A mainly originates from the southeast coast of China. As mentioned above, this part of the air mass mainly contains polluting fine aerosol loading. Sector C particles had the smallest $\tau$ ($0.14 \pm 0.11$) and $\alpha$ ranging from 0.15 and 1.95. This means that part C mainly included clean aerosols, with a relatively uniform distribution of coarse and fine particles. Sector B includes the China-Indochina Peninsula and the SCS. In this region, the possible sources produce continental and, to a lesser extent, marine aerosols. The optical depth value of part B was lower than that of the polluting source of part A and higher than that of the cleaning source of part C. In part B, the central value ($x_c$) of $\alpha$ was smaller than 1, indicating that marine aerosols bring about the significant growth of coarse particles. It must be emphasized that different aerosol types may be present in the air column simultaneously, influencing the observed optical parameters.

Interestingly, the relationship between the aerosol optical depth and wavelength index in different parts of the Taiping area has a similar distribution to that in the Dongsha area (Figure 9(a2,b2)). Similar to sector A of the Dongsha area discussed above, some of these cases are due to the intrusion of anthropogenic aerosols in the southwest, which produced high aerosol loading. In Part B, due to the addition of marine aerosols, the loading of coarse particles increases, and in Part C, there are mainly clean aerosols.

In summary, (1) the pollution aerosol in the Dongsha area mainly originates from the southeast coast of China, which mainly contains some fine particles; (2) the mixing of marine aerosols in the southwest of the Dongsha area increased the proportion of coarse particles; (3) the pollution aerosol in the Taiping area mainly originates from the area between Indonesia and Singapore; (4) The source of marine aerosol on Taiping Island may be more from the northeast direction; and (5) a small number of clean aerosol sources enter from the southeast of the two places, and the coarse and fine particles of this part of the aerosol were evenly distributed.

### 3.3. Seasonal Properties of Aerosol over Dongsha Island and Taiping Island

Monthly mean values of $\tau$ (500 nm), $\tau^F$ (500 nm), $\tau^C$ (500 nm), $\alpha$ (440–870 nm), FMF and their corresponding standard deviations in Dongsha and Taiping from January 2018 to December 2020 are shown in Figure 10a,b. The monthly average value is derived from the daily average data. The large values of $\tau$ (500 nm), indicating the high aerosol load,

were observed in March-April, while the minimum (0.093–0.096) was observed in June-July (Figure 10($a_1$)). The monthly value of $\alpha$ (440–870 nm) was always approximately 1, and all mean values of FMF were higher than 0.5, reflecting a relatively high fine particle loading in each month, especially in spring and autumn (Figure 10($b_1$)). For summer (May-August), the monthly mean values of mean $\alpha$ (440–870 nm) gradually decreased from 1.17 to 0.87, and the FMF decreased from 0.62 to 0.52, indicating an increase in coarse particle contribution during these months (Figure 10($b_1$)). It is also worth noting that all $\tau^F$ (500 nm) showed a pronounced decrease in summer, suggesting decreased loads of fine particles (Figure 10($a_1$)).

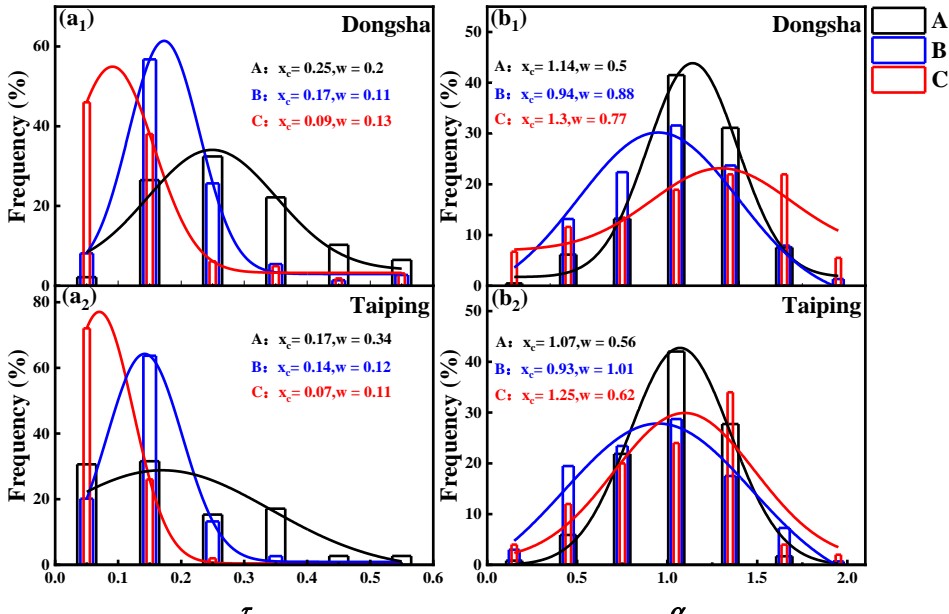

**Figure 9.** Distribution of the frequency of occurrences of ($a_1,a_2$) $\tau$, and ($b_1,b_2$) $\alpha$ in three sectors: A, B, C. These data are shown in black (sector A), blue (sector B) and red (sector C). The distribution of different sectors (colored lines) was identified by Gaussian fitting (fitted line, y = y0 + (A/(w*sqrt(pi/2)))*exp(−2*((x − xc)/w)^2)). The center value (xc) and waist width (w) of each part of the Gaussian function are shown.

Figure 11 shows the monthly wind direction probability obtained at Dongsha Island and Taiping Island from 2018 to 2020. In summer (May–August), the south wind prevails in the Dongsha area, while the northeast wind prevails in other seasons. In fact, northeasterly winds will bring more anthropogenic pollution sources (OC, BC, etc.) and dust aerosols, while in summer, there were no significant pollution intrusions. Additionally, the summer monsoon and southwest ocean current over the SCS (Figure 12) during this season may favour the accumulation of marine aerosols that can explain the coarse particle loads during summer in comparison with spring and autumn. In addition, the aerosols show lower fine particle loading and higher coarse particle loading in winter (Figure 10($a_1$)). At the same time, the monthly mean $\alpha$ (440–870 nm) of aerosols in winter was not more than 1, and the mean FMF value was approximately 0.5, indicating that the Dongsha area in winter has a higher loading of coarse particles than that in other seasons. In this sense, the high mass of sea salt aerosols observed in winter was associated with the increase in the significant height of combined wind waves and swell (SWH) (see, for example, Figure 10($c_1$)). The low aerosol loads registered in winter can be explained by the (Figure 10($c_1$)) high sea salt aerosol loads and the absence of pollution intrusions in this period. Figure 10($a_2,b_2,c_2$) shows the monthly mean values of $\tau$ (500 nm), $\tau^F$ (500 nm) and $\tau^C$ (500 nm) as well as $\alpha$ (440–870 nm) and FMF with the corresponding standard deviations at Taiping from January 2018 to December 2020. The largest values of $\tau$ (500 nm) were observed during August-September, while the lowest values (0.06–0.08) were measured from October to January (Figure 10($a_2$)). According to the

conclusion in Section 3.1, the high aerosol loading in August and September was mainly due to the high polluting aerosol brought by the southwest air mass. In addition, the pollution transport from the middle sea area surrounded by Singapore and Indonesia may be brought with the southwest wind at Taiping Island (Figure 11). There is no doubt that in winter, strong waves bring more sea salt aerosols in the Taiping area and the Dongsha area, making the aerosol optical depth at low values. Therefore, we can conclude that the unique meteorological characteristics and large-scale circulation in the South China Sea bring about different aerosol sources in the surrounding waters and then drive the aerosol properties of Dongsha Island and Taiping Island to show obvious monthly variation characteristics.

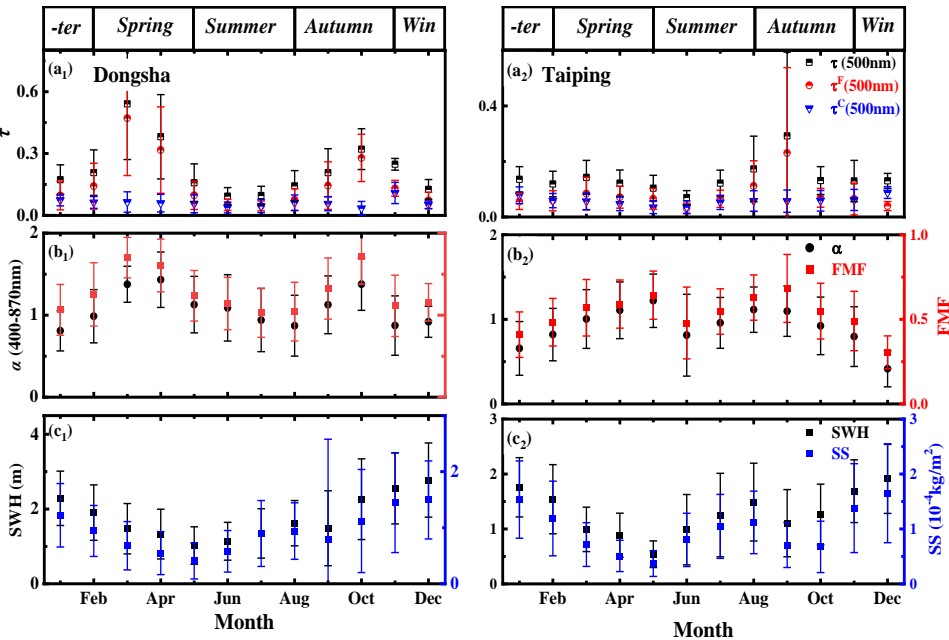

**Figure 10.** Monthly mean values of ($a_1$,$a_2$) $\tau$ (500 nm), $\tau^F$ (500 nm), $\tau^C$ (500 nm), ($b_1$,$b_2$) $\alpha$ (440–870 nm), ($c_1$,$c_2$) the significant height of combined wind waves and swell (SWH) and sea salt aerosol mass concentration FMF in Dongsha and Taiping from January 2018 to December 2020. The error bars are standard deviations.

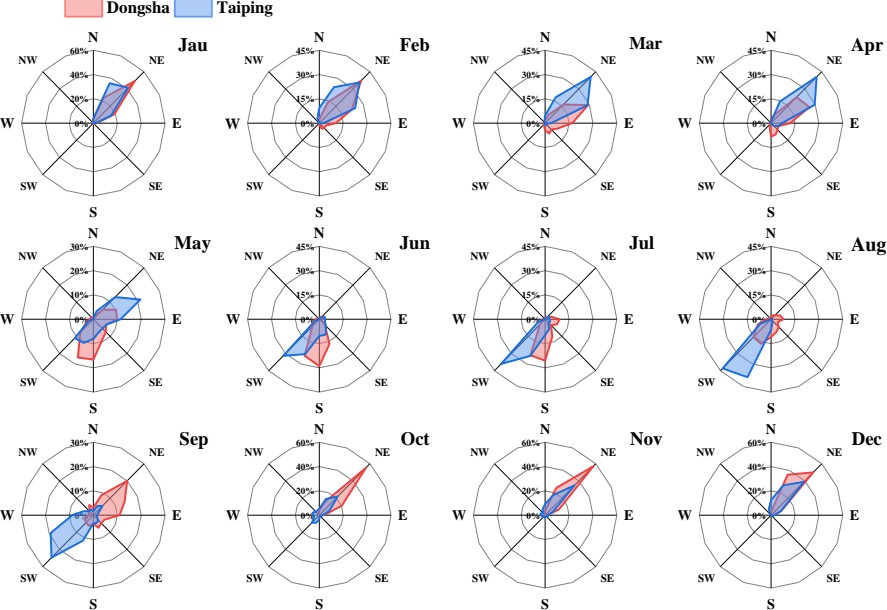

**Figure 11.** Monthly wind rose graphics obtained at Dongsha and Taiping island for three years.

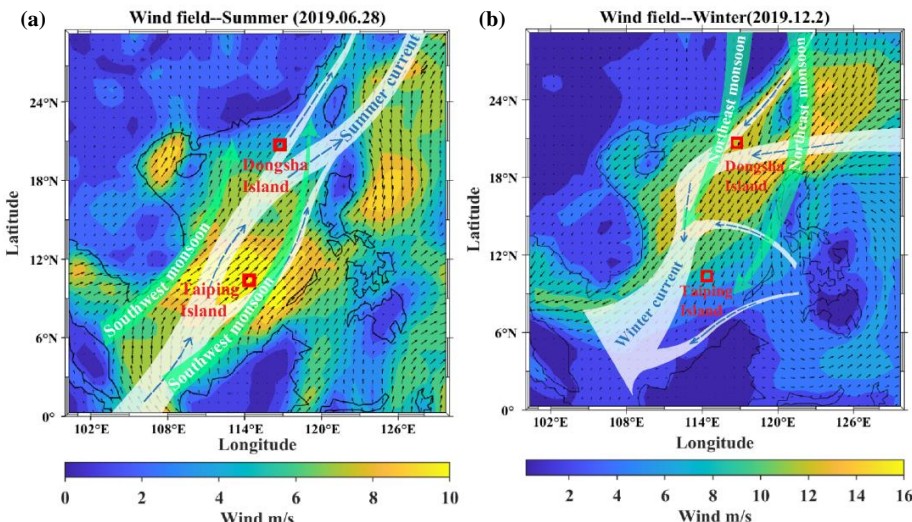

**Figure 12.** Distribution of wind field, monsoon and ocean current in the SCS. (**a**) Summer (**b**) Winter.

### 3.4. Regional Properties of Aerosol over the South China Sea

In this study, the AERONET data of Dongsha Island, Taiping Island and their surrounding four AERONET stations (see Figure 1) from 1 January 2018 to 31 December 2020 were used to study regional properties of aerosol over SCS (Figure 13). To analyze aerosol regional variability, we compared nearby sites using only time-consistent aerosol data (Table 3). Figure 13a shows comparisons of the daily average τ (500 nm) at Dongsha Island and Hong Kong Island from 1 January 2018 to 31 December 2020. Hong Kong is about 315 km from Dongsha Island. The temporal variation in the daily mean of τ (500 nm) was similar for both sites on most days during the analysis period, indicating that the processes controlling aerosol loading at both sites were similar. The correlation coefficient R between these two sites is 0.87. However, there are also large differences on some days (for example, τ (500 nm) at Dongsha Island was 0.05 on 1 July 2019, compared to 0.22 in Hong Kong).

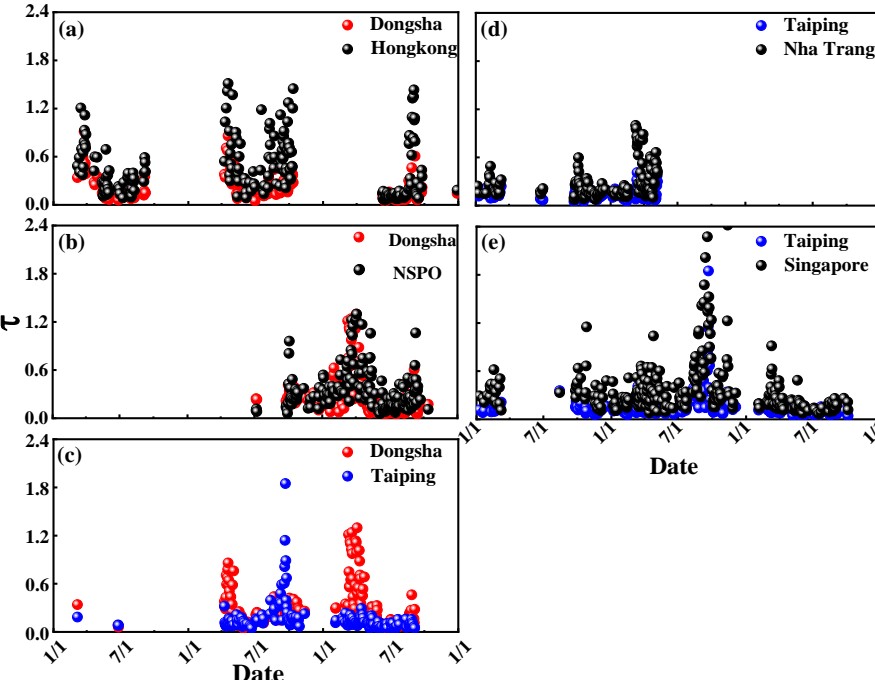

**Figure 13.** Comparisons of τ (500 nm): (**a**) Dongsha Island and NSPO, (**b**) Dongsha Island and Hong Kong, (**c**) Dongsha Island and Taiping Island, (**d**) Taiping Island and Nha Trang, and (**e**) Taiping Island and Singapore.

**Table 3.** Mean value of $\tau$, $\alpha$ (440–870 nm), $\tau^F$ (500 nm), $\tau^C$ (500 nm) and FMF obtained over Dongsha (DS), NSPO (NS), Hong Kong (HK), Taiping (TP), Nha Trang (NT) and Singapore (SP). Here, the same time series of stations are compared with Dongsha Island and Taiping Island from 1 January 2018 to 31 December 2020.

|  | DS | HK | DS | NS | DS | TP | TP | NT | TP | SP |
|---|---|---|---|---|---|---|---|---|---|---|
| $\tau$ (1020 nm) | 0.09 | 0.17 | 0.11 | 0.13 | 0.13 | 0.08 | 0.08 | 0.12 | 0.09 | 0.14 |
| $\tau$ (500 nm) | 0.21 | 0.44 | 0.25 | 0.35 | 0.31 | 0.16 | 0.14 | 0.27 | 0.17 | 0.37 |
| $\tau$ (380 nm) | 0.29 | 0.56 | 0.33 | 0.46 | 0.41 | 0.19 | 0.18 | 0.38 | 0.21 | 0.51 |
| $\alpha$ | 1.17 | 1.27 | 1.02 | 1.25 | 1.14 | 0.99 | 0.90 | 1.20 | 0.94 | 1.31 |
| $\tau^F$ | 0.15 | 0.37 | 0.18 | 0.31 | 0.24 | 0.10 | 0.08 | 0.21 | 0.10 | 0.30 |
| $\tau^C$ | 0.06 | 0.06 | 0.06 | 0.04 | 0.06 | 0.05 | 0.06 | 0.06 | 0.06 | 0.06 |
| FMF | 0.66 | 0.77 | 0.63 | 0.8 | 0.68 | 0.57 | 0.52 | 0.69 | 0.55 | 0.74 |
| Coincident day | 212 | 212 | 197 | 197 | 210 | 210 | 154 | 154 | 310 | 310 |

Differences in the timing and intensity of anthropogenic aerosol intrusion into these sites are most likely responsible for the differences in aerosol loading in these regions. In fact, the correlation in $\tau^F$ (500 nm) between Dongsha Island and Hongkong, R, is 0.82. The aerosol properties in the Dongsha area and NSPO area were also compared, and similar high correlation results were obtained (R = 0.73 for $\tau^F$ (500 nm)). From long wavelength (1020 nm) to short wavelength (380 nm), the difference between Dongsha Island and the other two places was becoming more and more obvious (Table 3). This is consistent with the previous conclusion that aerosol loading in the Dongsha area is mainly affected by fine particles in the southeast coast of China. On the other hand, for Taiping Island, the variation of optical depth in Taiping island was similar to those in Nha Trang (Figure 13d) and Singapore (Figure 13e). There is no doubt that the South China Sea region shows a similar coarse particle loading, and the coarse particle optical depth of the whole study area was maintained at about 0.06. In addition, the aerosol loading in Taiping area is significantly weaker than that in Dongsha Island, but the proportion of coarse particles (FMF = 0.57) was significantly higher than that in Dongsha Island (FMF = 0.68). Thus, we can conclude that the aerosols in the South China Sea were greatly affected by the aerosols in the surrounding urban areas, and the variation was very consistent with surrounding urban areas, but the aerosol loading was far lower than that in the coastal cities.

## 4. Conclusions

With unbalanced socio-economic development and a complex meteorological system, the aerosol pollutants in the South China Sea are of various types. To investigate the change of columnar aerosol properties over time and space in this little-studied region, aerosol optical depths obtained over Dongsha Island, Taiping Island, and four other nearby sites in the SCS were analysed. Within the analysed period, the mean of observed $\tau$ (500 nm) values over Dongsha Island and Taiping Island were significantly higher than those reported in the open ocean unaffected by long-range aerosol transport. The high aerosol loads over Dongsha were mainly linked to anthropogenic fine particle transport from the southeast coast of China and the occasional advection of desert dust from Mongolian areas. The high fine aerosol loading in the Taiping area originates from the southwestern region between Singapore and Indonesia.

The intrusion of aerosol particles from the southeastern coast of China, Singapore and Indonesia has brought high loading to the aerosols in the SCS. In particular, in the Dongsha area, 46% of the air masses come from the southeast coast of China, 36% from the China-Indochina Peninsula and the South China Sea, and 17% from the Luzon Strait and the SCS. Similarly, the Taiping area was also disturbed by various sources of air masses. Twenty-one percent of the air masses come from the area enclosed by Singapore and Indonesia, 55% from the Philippines, and 10% from the Sulu sea and the SCS.

The obvious seasonal cycles of aerosol optical depth and Ångström exponent over the SCS are caused by the seasonal distribution of meteorological regime over the SCS and

the mechanism of aerosol sources. In the Dongsha area, particles originating from spring show a maximum in aerosol optical depth, while in the Taiping area, aerosol particles from autumn show a higher optical depth. The seasonal increase of aerosol optical depth was related to the seasonal ocean current and wind direction in the SCS. In addition, the high wave height in winter in the two places produces the most sea salt aerosols, resulting in the smallest wavelength index and the lowest proportion of fine particles in Dongsha and Taiping. Aerosols in the SCS were greatly affected by the aerosols in the surrounding urban areas, and the variation was very consistent with surrounding urban areas, but the aerosol loading was far weaker than that in the coastal city.

**Author Contributions:** J.C.: Methodology, Formal analysis, Validation, Writing. W.Z.: Conceptualization, Resources, Supervision, Formal analysis, Funding acquisition. Q.L.: Conceptualization, Resources, Supervision, Formal analysis, Funding acquisition. X.Q.: Formal analysis, Writing—review & editing. X.C.: Funding acquisition, resources. J.Z.: Validation, Writing—review & editing. T.Y. (Tao Yang): Validation, Software. Q.X.: Validation, Software. T.Y. (Tengfei Yang): Validation, Software. All authors have read and agreed to the published version of the manuscript.

**Funding:** This research was funded by National Natural Science Foundation of China (Grant No. 41805014), Foundation of Advanced Laser Technology Laboratory of Anhui Province (Grant No. 20191002), Key Program in the Youth Talent Support Plan in Universities of Anhui Province (Grant No. gxyqZD2020032), Strategic Priority Research Program of Chinese Academy of Sciences (Grant No. XDA17010104), Key Laboratory of Science and Technology Innovation of Chinese Academy of Sciences (Grant No. CXJJ-21S028), Open Research Fund of Key Laboratory of Atmospheric Optics, Chinese Academy of Sciences (Grant No. JJ-19-01) and the President's Fund of Hefei Institutes of Physical Science (Grant No. YZJJ2021QN08).

**Data Availability Statement:** The data presented in this study are available on request from the corresponding author.

**Conflicts of Interest:** The authors declare that they have no known competing financial interests or personal relationships that could have appeared to influence the work reported in this paper.

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
