# Peer review of "Temporal Evolution and Regional Properties of Aerosol over the South China Sea"

_remotesensing, doi:10.3390/rs15020501_

Round 1
Reviewer 1 Report
I am glad to see that the authors of the article “Temporal evolution and Regional Properties of Aerosol over the South China Sea” conducted a study of the temporal evolution of aerosol properties in the South China Sea, a little studied region.
The article brings an extensive study of these properties and presents a large amount of statistical results. However some corrections and clarifications are needed, which I will present bellow:
Abstract:
line 24: complicated meteorology system – complex meteorological system
Line 26: meteorology law: I would suggest: meteorological/ climate regime
Line 29: great affection – The word affection is not suited fot this case, what is the meaning of the phrase?
Introduction
Line 47: this aera – this area
Line 55: understending the function of aerosols in Earth’s radiation – Earth’s radiation budget.
Line 56: Through Spatial – I suggest From sattelite and ground remote…
Section 2:
Line 126: is incorporated – are incorporated
145 : I suggest to cite HYSPLIT.
Section 3
From my point of view, is missing a clear definition of how was calculated α and measured τ and FMF.
Lines 157 and 159: I do not recommend the use the expressions: “most facinating thing” and “to get an idea”.
Line 162: From my point of view, you should use α instead of τ to explain your argument, anyway, a citation should be necessary here.
Table 1 and 2 : What are Min, Med, Max colunms? There is no information about it.
It is not clear from tables 1 and 2 the values obtained for τ and α discussed at lines 157 to 172. Some of values shown in the text can not be found at the table. The text needs clarification.
A better explanation of these numbers will clarify the analysis done.
At the abstract you use α(380-870 nm) but at the text, the value for 380 nm does not appear.
The reference cited at line 193 (Smirnov et al) is missing in the reference section.
Line 233: Which criteria was used to define that the high aerosol load days were those with τ> 0.35? May be some citation is needed here.
Line 240: A verb is missing in the phrase.
Line 300: No mention to sector D. May be you also could define the sector D at this line.
Lines 302 to 306: Sentences are being repeted.
Fig. 9 The gaussian curves parameters could help clarify the discussion.
Line 325: It is not clear from Fig.9 that 80% of alpha values from A (dongsha) are higher than 1. Again, if we had the parameters would be easier to calculate this value.
Line 333: It is not clear what do you mean in the sentence: “The α value of part B increases significantly while α is smaller than 1” – it needs clarification.
Line 337: The same is need for the sentence: “Interestingly, the relationship between the aerosol optical depth and wavelength index in different parts of the Taiping area has a similar distribution to that in the Dongsha area (Fig. 9 a2, b2).”
Which relationship do you mean?
Line 339: above is mispelled.
Line 396: “We can concluded” – “We can conclude”
Line 430: a verb is missing here.
line 447: Sophisticated is not the ideal word for classifying the meteorological system, do you mean complex?
Line 455: Mongolia is mispelled – should be Mongolian areas?
line 466 : “Seasonal distribution of meteorological over” - missing word.
Reviewer 2 Report
Review of "Temporal evolution and Regional Properties of Aerosol over the South China Sea" by Jie Chen et al.
Recommendation: Major Revisions
The manuscript "Temporal evolution and Regional Properties of Aerosol over the South China Sea" mainly studies the temporal evolution properties of aerosols and their regional characteristics based on the three-year observations from AERONET in the South China Sea. In general, the paper is well written and presented in a logical way. It is a timely and important piece of work, and of general interest for aerosol properties and seasonal cycle over South China Sea related communities. I therefore recommend publication of this paper in Remote Sensing after major revisions. My comments are listed as follows:
Major Comments:
1. It is suggested that more and detail background should be added into the Introduction part. In this manuscript, some studies have been given, but these seem to be not very relevant to this study, and also not sufficient to help us understand how this study differs from previous studies.
2. According to the AERONET homepage, the observation has been operated at Dongsha Island and Taiping Island sites over the years. So, why were the three-year observation only used in this study?
3. Authors defined the high aerosol loads if the AOD at 500nm is greater than 0.35. So, how to determine this value?
Specific Comments:
1. Line 214 and 221: “340nm” in Table 1 and Table 2 should be replaced by “380nm”
2. Line 229: “(380-870nm)” should be replaced by “(440-870nm)”
Round 2
Reviewer 2 Report
My comments have been addressed. So, there is no any comments for this paper.